# Fault Detection in 3D Printing: A Study on Sensor Positioning and Vibrational Patterns

**DOI:** 10.3390/s23177524

**Published:** 2023-08-30

**Authors:** Alexander Isiani, Leland Weiss, Hamzeh Bardaweel, Hieu Nguyen, Kelly Crittenden

**Affiliations:** Mechanical Engineering, College of Engineering and Science, Louisiana Tech University, Ruston, LA 71272, USA; aci004@latech.edu (A.I.); lweiss@latech.edu (L.W.); hamzehb@latech.edu (H.B.); htn005@latech.edu (H.N.)

**Keywords:** additive manufacturing (AM), Fused Filament Fabrication (FFF), nozzle clogging, fault detection

## Abstract

This work examines the use of accelerometers to identify vibrational patterns that can effectively predict the state of a 3D printer, which could be useful for predictive maintenance. Prototypes using both a simple rectangular shape and a more complex Octopus shape were fabricated and evaluated. Fast Fourier Transform, Spectrogram, and machine learning models, such as Principal Component Analysis and Support Vector Machine, were employed for data analysis. The results indicate that vibrational signals can be used to predict the state of a 3D printer. However, the position of the accelerometers is crucial for vibration-based fault detection. Specifically, the sensor closest to the nozzle could predict the state of the 3D printer faster at a 71% greater sensitivity compared to sensors mounted on the frame and print bed. Therefore, the model presented in this study is appropriate for vibrational fault detection in 3D printers.

## 1. Introduction

Recently, additive manufacturing (AM) has undergone rapid evolution and is now regarded as a critical technology in the Fourth Industrial Revolution (Industry 4.0) [1,2,3,4,5,6,7]. AM has been integrated into manufacturing and production due to its contribution to minimizing cost, waste of fabrication material, and time of production of complex parts [8,9,10]. AM makes it economically feasible to design and mass-produce products in a timely fashion [10]. In customizing and fabricating complex 3D geometric prototypes, Fused Filament Fabrication (FFF) techniques, also known as 3D printing which is widely used and one of the fast-growing AM technologies, are being used daily across several fields including education [11,12,13,14,15,16,17], biomedical [12,18,19,20,21], medical [9,10,11], robotics [11,22,23,24,25,26], food [24,27,28,29,30], engineering [15,16,21], aerospace [9], etc. There is a rapid demand for 3D printing in different fields and across the board [12,30,31,32]. However, there is a need to mitigate the imminent challenges of an estimated 20% printing failure rate when unskilled users operate FFF and the lower reliability when compared with other forms of AM processes [12]. Some of these challenges are due to nozzle clogging or blockage [11,12,32,33,34], qualifying the printed parts [8,35,36,37], quality control [12], and detecting failed prints during the fabrication process [8,9,12]. Studies have proven that the two main purposes for in situ monitoring of FFF processes are to ascertain the machine’s health status and the quality of the building parts [9,12,18,37,38,39,40]. Therefore, it is important to become aware of and diagnose the state or condition of a 3D printer to make necessary adjustments or even replace worn parts [3,22,38,41].

With the use of sensors [41], vast contributions have been made by researchers for reliable in situ monitoring, and several approaches to mitigate unwanted vibrations caused by unintended movements in the 3D printer, which often result in loss of product quality, deterioration, and damage of the machine [22,42], have also been developed. The acoustic emission technique was used [3] to determine the machine’s condition. Li et al. [9] used vibration sensors to detect normal and filament jam states of an FFF machine, and to predict the product quality during the FFF process using two data-driven methods—the least squares support vector machine (LS-SVM) and the back-propagation neural network (BPNN); however, the filament jam was not studied at various printing or extruding temperatures. Similar fault diagnosis methods were also used by Kun et al. [43]. Guo et al. [23] used the Transfer Support Vector Machine (TSVM) technique for fault diagnosis. Jhodkar et al. [8] created artificial conditions for nozzle clogging by reducing the print temperature of polylactic acid (PLA) which gradually altered the layer thickness and observed the vibrational change in amplitude through an accelerometer. Tlegenov et al. [11] also monitored the nozzle condition in 3D printing but in a more fascinating approach through a gradual reduction of the nozzle extrusion temperature of PLA, acrylonitrile butadiene styrene (ABS), and SemiFlex filaments for both direct and bowden types of FFF extruders with an accelerometer. The changes in amplitude observed in [8,11] were not qualitatively analyzed, and using the Fast Fourier Transform (FFT) can be challenging when used to distinguish between the normal and clogged state of the 3D printer because both states can occur at a similar frequency. Zhang et al. [31] optimized the vibrational characteristics of a colored 3D printer using finite element modal (FEA) analysis to obtain the natural frequency of colored 3D printers and compared it with the mechanic’s theoretical dynamic characteristics. Using frequency domain analysis, the power spectrum of the vibrational signal was analyzed and the resonance position of the 3D printer was obtained by comparing the experimental analysis with FEA. Chuan et al. [22] used a deep learning method, Fusing Convolutional Generative Adversarial Encoders (FCGAE), to detect faults with only normal condition signals. Zhang et al. [30] used an acceleration sensor-based vibration test to explore the relationship between clogging and vertical vibration characteristics of the nozzle under different feed parameters. They affirmed that higher feed torque can be used to improve the feed speed and eventually improve the quality of color mixing nozzle 3D printers but using vibration acceleration amplitude alone to quantify the difference in vibration during blockage is not clear enough since both normal and abnormal states of printing occur at the same frequency. Rao et al. [44] identified failure mode and anomalies in FFF using Bayesian non-parametric analysis and Evidence Theory for in situ heterogeneous sensor data—thermocouples, accelerometers, infrared temperature sensor, and a real-time miniature video borescope. Gomathi et al. [45] did not explicitly analyze or propose any condition-monitoring technique after stating the three obvious printing motions (continuous, point-to-point, and zig-zag motion) observed during the operation.

While a few studies have employed machine learning, an approach to check the vibrational fault induced on the nozzle, frames of the printer, and the print bed has not been developed. The use of FFT has been found to be primarily employed to analyze the vibrational signals, but limitations in its ability to provide a comprehensive analysis and effectively classify faults have been observed. To address this, a novel approach has been presented that combines multiple techniques, including FFT, spectrogram, Principal Component Analysis (PCA), and Support Vector Machine (SVM), to comprehensively analyze vibrational signals obtained from different printing conditions and identify and classify faults. The results of this study have demonstrated that the proposed approach is more effective in identifying and classifying faults than prior methods, thus addressing the research gap and providing a valuable contribution to the field of vibrational analysis during fabrication.

## 2. Materials and Methods

The printer and sensors in this work were chosen for their general availability and applicability. The ADXL335 accelerometer was chosen for this study due to its high sensitivity and ability to measure acceleration across three axes, making it ideal for capturing the complex vibrational patterns associated with 3D printing. Furthermore, its signal-conditioned voltage outputs allow for easy integration with data acquisition tools like LabVIEW. On the other hand, the Lulzbot Mini 3D printer was selected due to its widespread use in both educational and professional settings, making our findings broadly applicable. Additionally, its open-frame design allows for easy placement and adjustment of sensors, facilitating a more comprehensive analysis of the printer’s operation.

Figure 1 shows the experimental setup that is used for in situ monitoring. In this work, three accelerometers with the model number ADXL335 were used to properly access, monitor, and measure the operating state of a Lulzbot Mini 3D printer by collecting operational vibration signals under different printing conditions. The ADXL335 is a complete 3-axis accelerometer with signal-conditioned voltage outputs, and it measures acceleration with a minimum full-scale range of ±3 g. Also, it has a typical sensitivity of 270 mV/g. These sensors were mounted in different positions (the nozzle, the frame, and print bed) on the Lulzbot mini 3D printer and connected through a National Instrument tool (NI 9202) to LabVIEW to collect the time-domain vibrational signals. These signals acquired were post-processed, analyzed, and visualized using MATLAB.

An overview of the sensor system can be seen in Figure 2. Sensor 1 was used to monitor the nozzle vibrational behavior during normal and clogged situations; Sensor 2 was used to observe corollary vibrational variation on the frame of the 3D printer. Sensor 3 was used to diagnose the vibration from both the nozzle and printed material on the print bed. As such, the third sensor was used for quality control and to detect wobbling of the fabricated sample.

To ensure that the sensors were working effectively and efficiently, confirmatory tests were conducted via a shaker table to ascertain the sensitivity of the accelerometers at a frequency of 200 Hz and gravity of 0.2 g, 0.4, and 1.3 g. The sensors were placed precisely on the shaker table to validate this test. The time-domain datasets obtained from the three sensors were converted to the frequency domain using Fast Fourier Transform (FFT). A sample of the data collected from Sensor 1 (on the nozzle) is shown in Figure 3. The other two sensors performed equally well. As indicated in the figure, the sensors were functional and could detect vibrations at the set frequency.

In these experiments, PLA (polylactic acid) was selected as the material for printing because it has minimal warping, produces little or no odor during printing, and has a lower melting point when compared to other filaments like ABS (acrylonitrile butadiene styrene). Additionally, it is non-toxic and safe to use. Both clogged and unclogged nozzles were studied in situ via the three sensors. A baseline of operation established for the normal FFF process was at 195 °C, 60 °C, and 60 mm/s for the extrusion temperature, print bed temperature, and feed rate, respectively. Abnormal or faulty FFF processes in the 3D prints were purposefully initiated for analysis. The nozzle was intentionally clogged by reducing the print temperature. This approach followed the clogging concept used unanimously by different researchers [8,11]. A sampling rate of 3.0 kHz was used to assure computational integrity and efficiency.

Table 1 shows the operating conditions of the printer during testing. The nozzle was allowed to transition from normal operation with an unclogged state into printer functions that produced increasingly unacceptable final print products. As extrusion temperature was reduced, printer function became more limited and eventually non-functional (completely unacceptable) as indicated. This set the progression of tests that were monitored by the sensors.

As the nozzle became more clogged, the sensors could accurately detect variations in the print state. The sensor mounted on the nozzle was the most sensitive because of the sensor’s proximity and direct contact with the nozzle carriage. Figure 4, Figure 5 and Figure 6 show the normalized acceleration–time graph of the sensors. The normalized charts were based on the raw voltage data collected by the sensors, converted to accelerations using the accelerometer’s sensitivity of 270 mV/g, and normalized with respect to the maximum value of Sensor 1. Based on the amplitude of acceleration obtained from the normalized graphs above, sensor one was roughly 71% more sensitive than the other two.

With the operating parameters set, two prototype print designs were investigated across the operating conditions. First, as shown in Figure 5a, a rectangular prototype was fabricated with the filaments extruding out of the nozzle onto the print bed at angles of 45 and 135 degrees. Second, an octopus prototype that was supplied with the 3D printer from Lulzbot Mini [45] was utilized (Figure 5b).

Over 300,000 data points were collected over a 6 min timeframe using LabView during the FFF process for each of the conditions shown in Table 1. To look for patterns in each dataset and to explore the information contained in it, spectrogram, Principal Component Analysis (PCA), and Support Vector Machine (SVM) approaches were taken to aid data visualization.

In the FFF process, frequencies with higher and lower amplitudes of vibration were identified. The spectrogram shows the spectrum of frequencies and visually displays the major frequencies brighter than the minor ones.

Using dimensionality reduction, the data were separated into groups based on shared characteristics. Principal Component Analysis (PCA) performed a linear transformation on the dataset and it was determined that most of the variance in each dataset was captured by the first and second principal components. Clustering or grouping the dataset with PCA determines the underlying structure and interprets the data based on input data. This gave us additional insight into the data collected more than using FFT alone. Using only FFT to determine the machine’s states was not enough to determine the change in vibration since it only shows a fraction of the hundreds of thousands of data points. Finally, a Support Vector Machine (SVM) was developed to classify normal and faulty regions. However, the print state at 185 °C (which is very close to the normal print state) was ignored during the dimensional reduction processes since the spectrogram showed little or no variation compared to the normal printing condition.

## 3. Results and Discussion

Using the Fourier amplitude spectrum of acceleration or Fast Fourier Transform (FFT), the dominant vibrational frequency and the corresponding amplitude for each printing temperature were observed, as shown in Figure 6a–c, for the different sensor positions.

Out of the three sensors mounted on the 3D printer, the sensor mounted close to the nozzle (Sensor 1) showed clearer variation than the other two. The amplitudes of vibration from the sensor mounted on the print bed (Sensor 3) were high because more vibrations were induced on the bed from the nozzle (which is forcing itself to extrude filaments), and the wobbling print. However, irrespective of the sensor position, the frequency range between 0 and 120 Hz was associated with the cooling fan located close to the nozzle, which is constantly on to hinder the melted thermoplastic on the heat block from fully backing into the heat sink.

A spectrogram model was employed to enhance the visualization of vibrational signals during various printing conditions. The spectrogram model generates the graph of the spectrum of frequencies vs. time for each dataset. Figure 7a–c present the two-dimensional comparison graphs for different printing conditions on the three sensors. The spectrogram graphs depict partially and wholly clogged operations represented by brighter regions, while darker regions represent the unclogged or slightly clogged printing condition. The duration of the comparison graph for each sensor was approximately 1500 s, with a duration of about 350 s for each dataset collection for different print states.

In order to optimize the analysis of sensor signals in a 3D printing system, we conducted a comparative study between spectrograms and Fast Fourier Transforms (FFT). Our objective was to discern which method would provide a more nuanced understanding of the variations in the signals, especially during different printing conditions. The findings from this comparison not only shed light on the superior method but also unveiled distinct vibration patterns during fault conditions in the 3D printer. The results indicated that the use of spectrograms revealed clearer variations in the signals from each sensor compared to the use of FFT. Further analysis of the spectrograms revealed changes in patterns at different print conditions over time. The spectrogram provided a more comprehensive view of the signal, which enabled the identification of patterns that were not visible through the use of FFT alone. Variations in the operating frequencies between normal and abnormal conditions were discovered through the analysis. The signals from the three sensors were analyzed and it was observed that the 3D printer exhibited distinct vibration patterns during fault conditions. Neglecting the frequency between 0 and 120 Hz, Table 2 shows the dominant frequencies in the different printing conditions.

To further analyze the vibrational signals under different printing conditions, a data-driven approach was employed utilizing Principal Component Analysis (PCA) and Support Vector Machine (SVM) algorithms. The datasets collected were utilized for clustering, training, and testing. The resulting model was then utilized to visualize the differences in vibration during normal and faulty operations. To reduce the complexity of the over 300,000 data points collected, dimensionality reduction was employed using PCA. This resulted in a reduction of the data points to 4000 points, utilizing the first two principal components that stored the maximum variation of the dataset. These dimensions were projected onto a 2D Cartesian plane, and to further demonstrate the distinction of clustering regions for normal and faulty conditions, the third principal component was included as a visual aid, as depicted in Figure 8, Figure 9 and Figure 10. To demonstrate the absence of overlap in normal and faulty clusters, we also employed 3D figures.

Based on the data visualizations presented, there was a distinct separation in the clustering patterns between the normal and unacceptable print states. Further analysis revealed that a common vibrational cluster was observed to occur during the fabrication of the contour or walls of the sample, and another distinct vibrational cluster was observed when fabricating the infill of the sample. However, the vibrational patterns obtained from the three sensors may not be identical, as this could be attributed to the specific nature of the vibrational signals being detected by each sensor based on where they were mounted on the 3D printer. This highlights the importance of considering the sensor placement when interpreting the results.

To facilitate the analysis of the data, a reduction in the number of data points from 4000 to 10 was implemented to capture the regions of major variations in vibration. This simplification of the clustering process also enabled the utilization of a Support Vector Machine (SVM) algorithm to classify the regions in which each in situ monitoring condition is likely to occur. As depicted in Figure 11, Figure 12 and Figure 13, the results of the segmented Principal Component Analysis (PCA) of the trained data and a combination of PCA and SVM applied to the test data for both normal and unacceptable print conditions are presented.

In the classification of printing conditions, the SVM algorithm finds the hyperplane that maximally separates the data points belonging to different classes, so that data points on one side of the hyperplane belong to a normal print, and those on the other side belong to unacceptable prints. Hence, the Support Vector Machine (SVM) was used to estimate the regions where normal and unacceptable vibrations will occur.

The same analysis was carried out with a stereolithography file of an octopus shape, with the same printing conditions maintained for normal and unacceptable prints (195 °C and 165 °C, respectively). Vibrational differences were observed between the two printing conditions using spectrograms, PCA, and the developed SVM tools. The spectrogram graphs from the three sensors unanimously agreed with the vibrational differences, showing a clear amplitude of vibration for normal print (Figure 14a–c). The frequencies of operation were well highlighted in the spectrogram of the normal print than in the faulty print. The spectrogram graphs of the faulty print condition showed little or no amplitude of vibration.

Neglecting the frequency between 0 and 120 Hz, Table 3 shows the dominant frequencies under different printing conditions. During the fabrication, it was observed that as the nozzle was completely clogged, there were no peculiar or dominant frequencies during operation. As the nozzle became completely clogged, numerous frequencies contributed to the vibration, and the spectrogram suggested the difficulty in revealing the dominant frequencies.

Subsequently, the collected data were processed using Principal Component Analysis (PCA) and Support Vector Machine (SVM) algorithms. The results of the analysis revealed the presence of differences in vibrational clustering across the three sensors. The predicted outcomes obtained through the application of PCA and SVM are presented in Figure 15, Figure 16 and Figure 17 below.

This study’s results indicate that distinct vibrational clusters were observed among the different samples. The applicability of the trained data obtained from a rectangular sample for testing data collected from an octopus print was limited. So, a normal print of an octopus shape was trained and used to test data for the acceptable and unacceptable print states.

Further analysis on the octopus print utilizing segmented Principal Component Analysis (PCA) and Support Vector Machine (SVM) revealed that Sensor 2, which was mounted on the frame, exhibited a level of error as some segment points were found to be in the same region as the unacceptable print condition. This discrepancy may be attributed to the sensor’s positioning and inability to accurately detect variations in the sophisticated print process. In contrast, Sensor 1 and Sensor 3, which were positioned close to the nozzle and the print bed, respectively, demonstrated clear distinctions between vibrational prints regardless of the shape of the sample being fabricated. These findings suggest that the detection of vibrational faults is facilitated by the positions of Sensor 1 and Sensor 3. However, Sensor 1 was the most effective among the three sensors in terms of sensitivity and detecting vibrational patterns. This implies that higher vibration levels were detected on the nozzle during clogging, as it exerts force to extrude the filament.

## 4. Conclusions

This paper has presented a quantitative analysis of the vibrational operation of Fused Filament Fabrication (FFF) printing for various printing states. Utilizing FFT, the three sensors captured the changes in vibrational frequencies, and different dominant frequencies were observed as the nozzle moved from an unclogged to completely clogged condition. However, as the printed prototype became complex, it was challenging to note the predominant vibration frequency when the nozzle was completely clogged. Based on the analysis, the important vibration frequencies are roughly 370 Hz and 270 Hz under the normal print condition. This vibration amplitude was detected by the three different sensors and at different degrees, with Sensor 1 being the clearest and most distinct. The vibration on the frame was minimal and unstable to be distinguished clearly by Sensor 2 because of the damped frame by Lulzbot. Sensor 3 had a different clustering pattern and vibration because the build plate movement is one-directional (*y*-axis movement). The information obtained from Sensor 3 can be used for quality control analysis and wobbling detection. Further research will focus on using vibrational patterns to detect z-banding and skewness effects in fabricated parts.

## Figures and Tables

**Figure 1 sensors-23-07524-f001:**
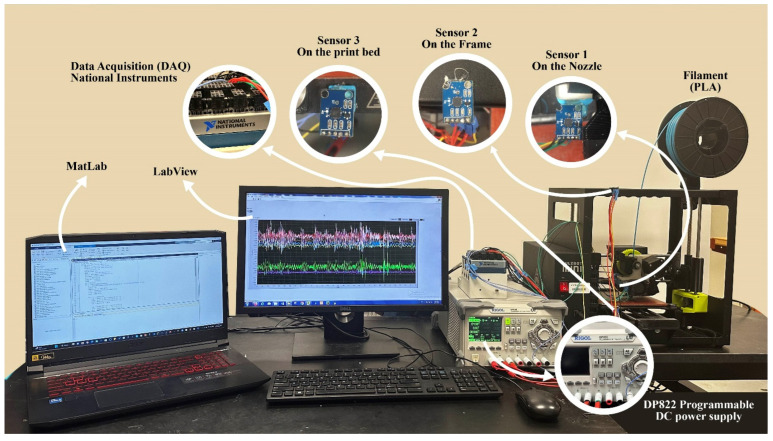
Lab experimental setup.

**Figure 2 sensors-23-07524-f002:**
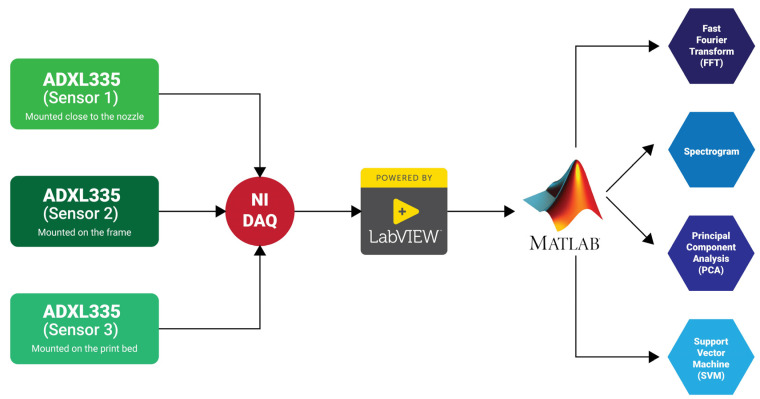
Schematic of the block diagram for the experimental setup.

**Figure 3 sensors-23-07524-f003:**
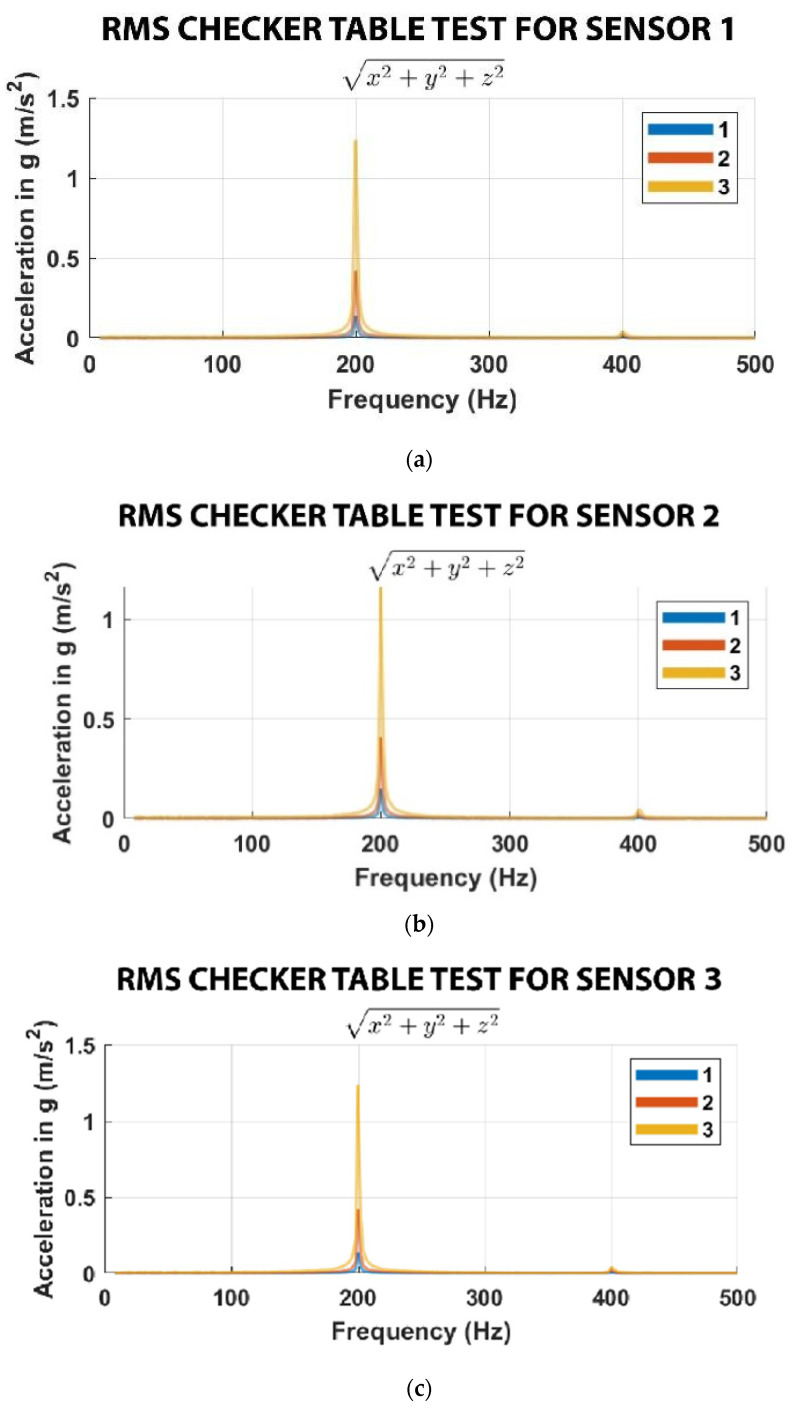
Sample of calibration data for (**a**) Sensor 1 (on the nozzle), (**b**) Sensor 2 (on the frame), (**c**) Sensor 3 (on the print bed).

**Figure 4 sensors-23-07524-f004:**
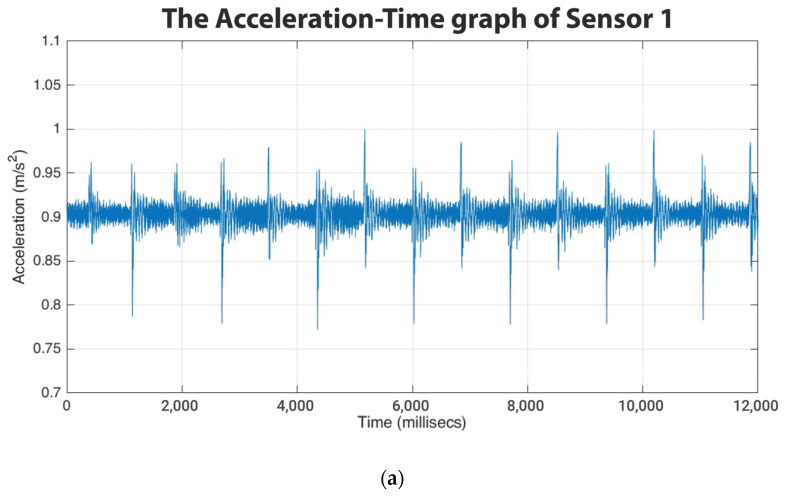
Normalized acceleration–time graph of (**a**) Sensor 1 (on the nozzle), (**b**) Sensor 2 (on the frame), (**c**) Sensor 3 (on the print bed).

**Figure 5 sensors-23-07524-f005:**
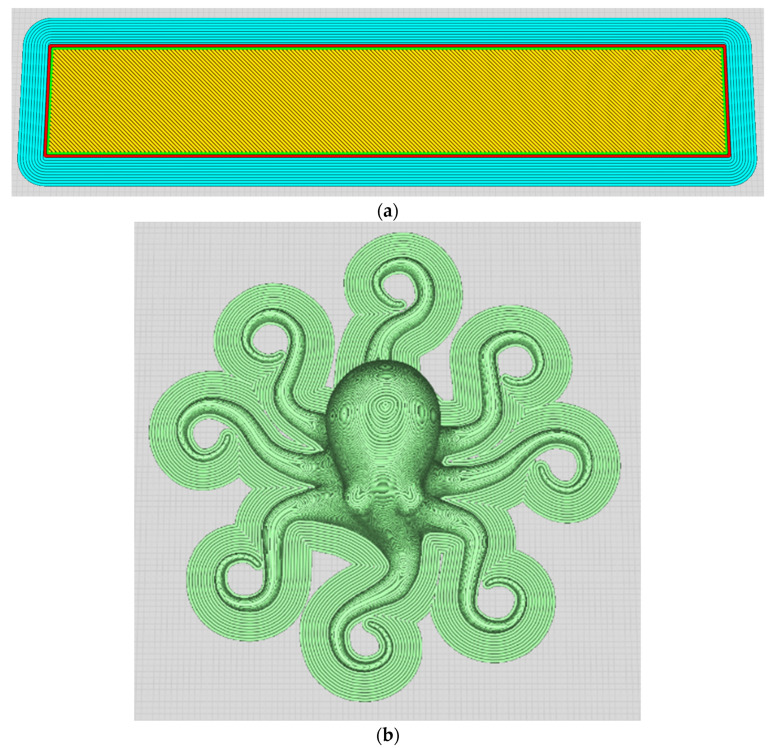
(**a**) Rectangular print sample. (**b**) Octopus print sample.

**Figure 6 sensors-23-07524-f006:**
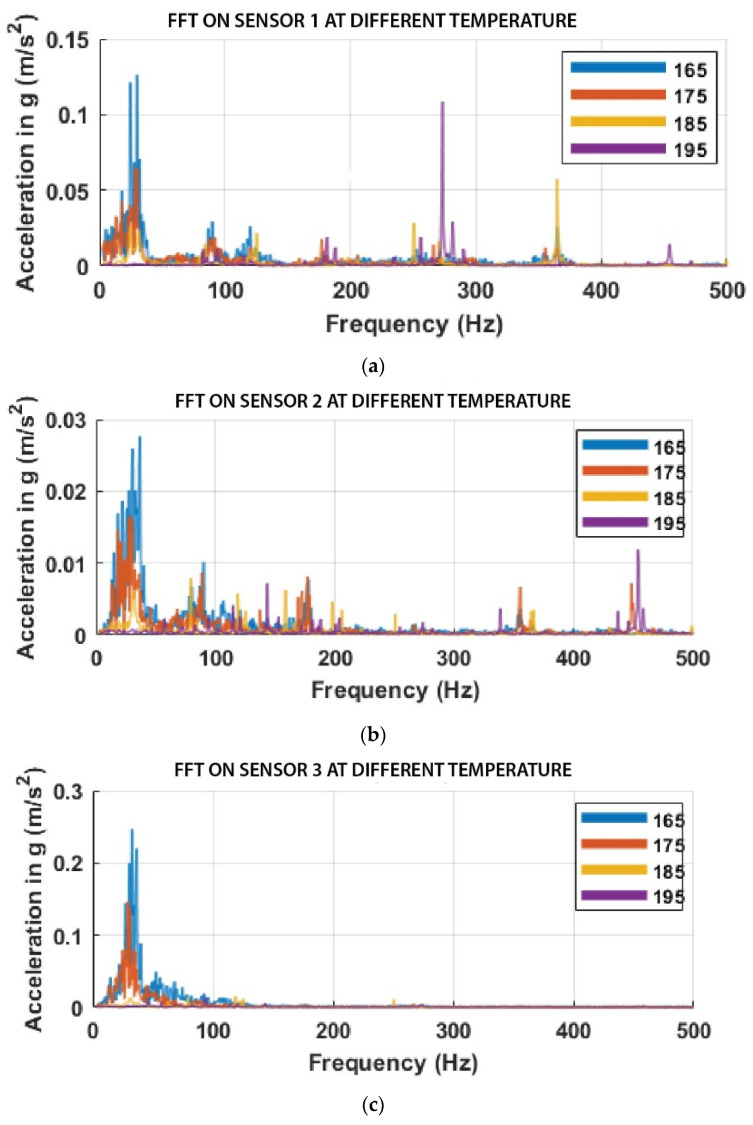
FFT at different printing temperatures based on the sensor positions: (**a**) Sensor 1, mounted close to the nozzle; (**b**) Sensor 2, mounted on the frame, and (**c**) Sensor 3 mounted on the print bed.

**Figure 7 sensors-23-07524-f007:**
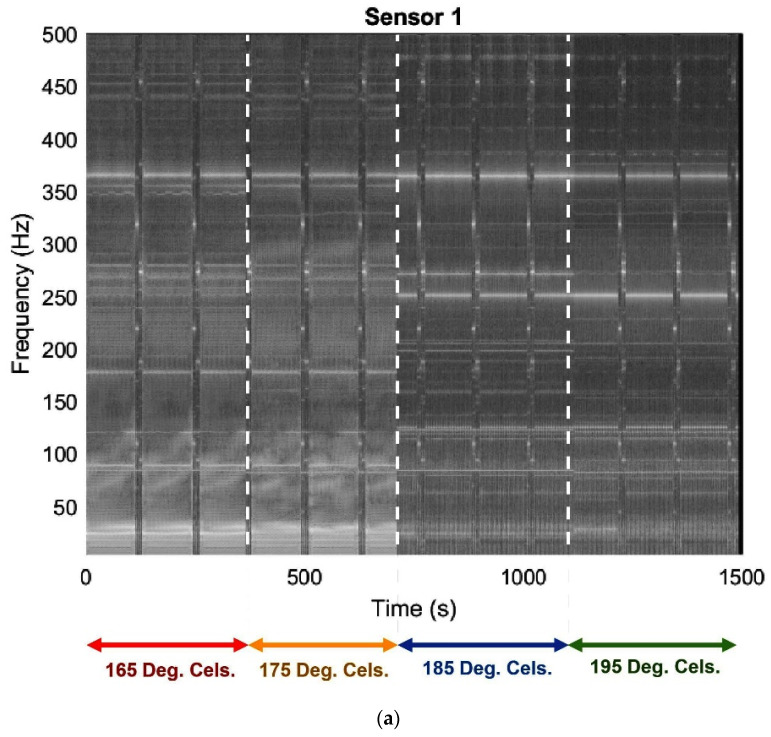
Spectrogram at different printing temperatures: (**a**) Sensor 1, mounted close to the nozzle, (**b**) Sensor 2, mounted on the frame, and (**c**) Sensor 3, mounted on the print bed.

**Figure 8 sensors-23-07524-f008:**
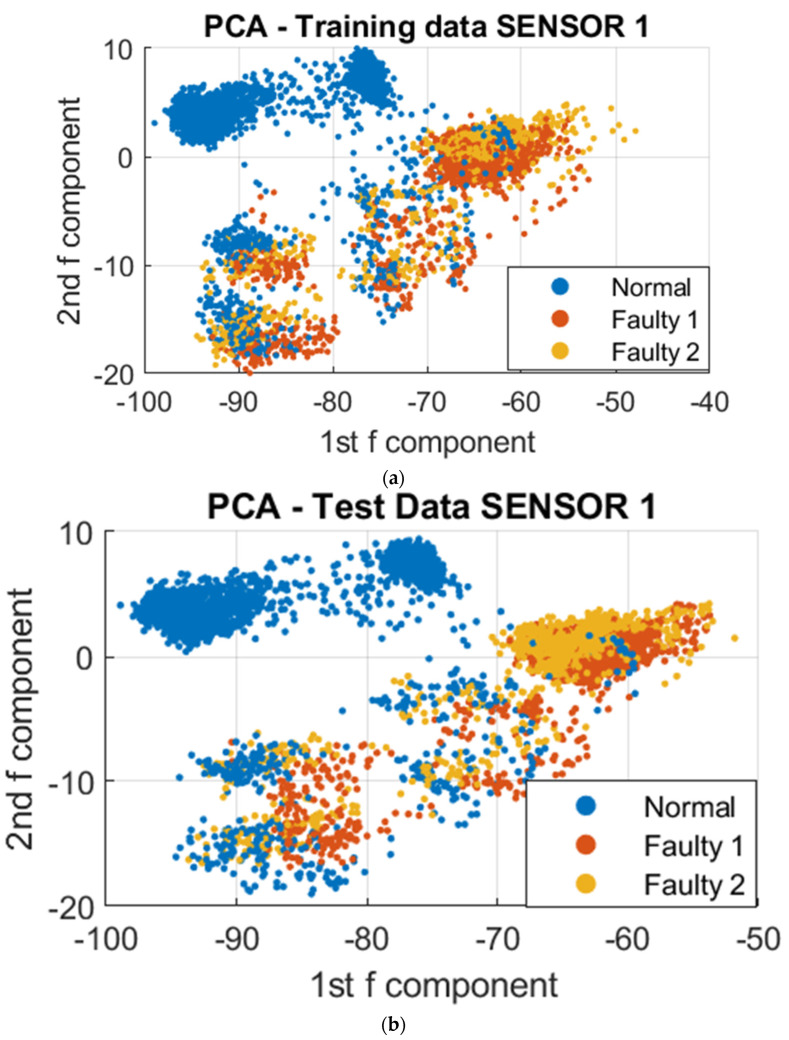
PCA on Sensor 1 at different print temperatures; Faulty 1 and 2 are the unacceptable conditions of 175 ° and 165 °, respectively. (**a**) 2D PCA for trained data; (**b**) 2D PCA for test data; (**c**) 3D PCA for trained data; (**d**) 3D PCA for test data.

**Figure 9 sensors-23-07524-f009:**
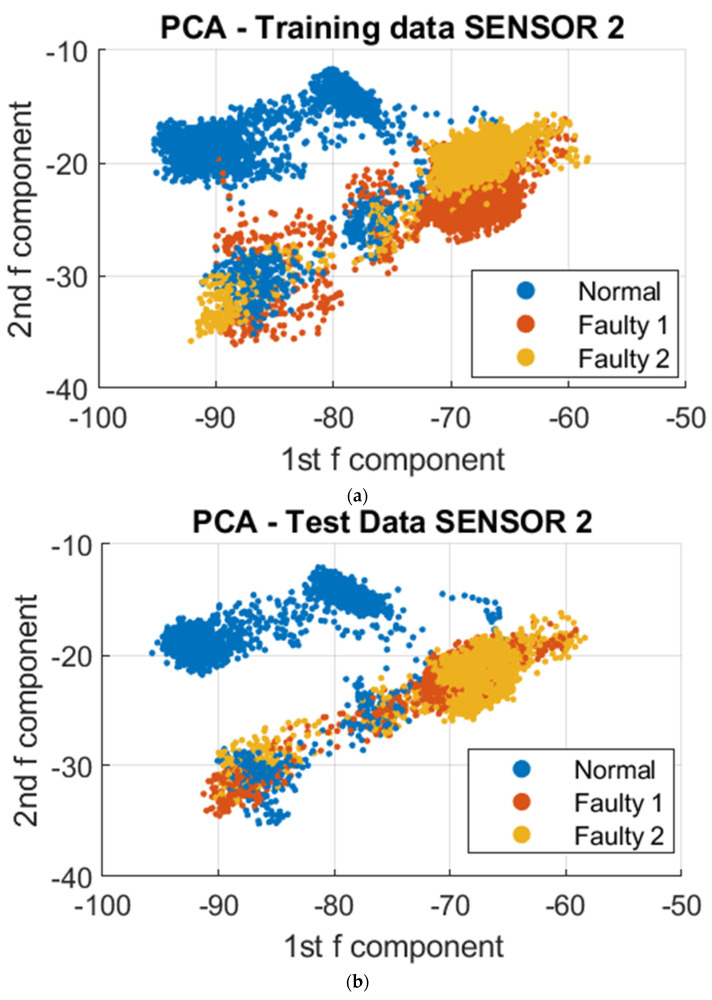
PCA on Sensor 2 at different print temperatures; Faulty 1 and 2 are the unacceptable conditions of 175° and 165°, respectively. (**a**) 2D PCA for trained data; (**b**) 2D PCA for test data; (**c**) 3D PCA for trained data; (**d**) 3D PCA for test data.

**Figure 10 sensors-23-07524-f010:**
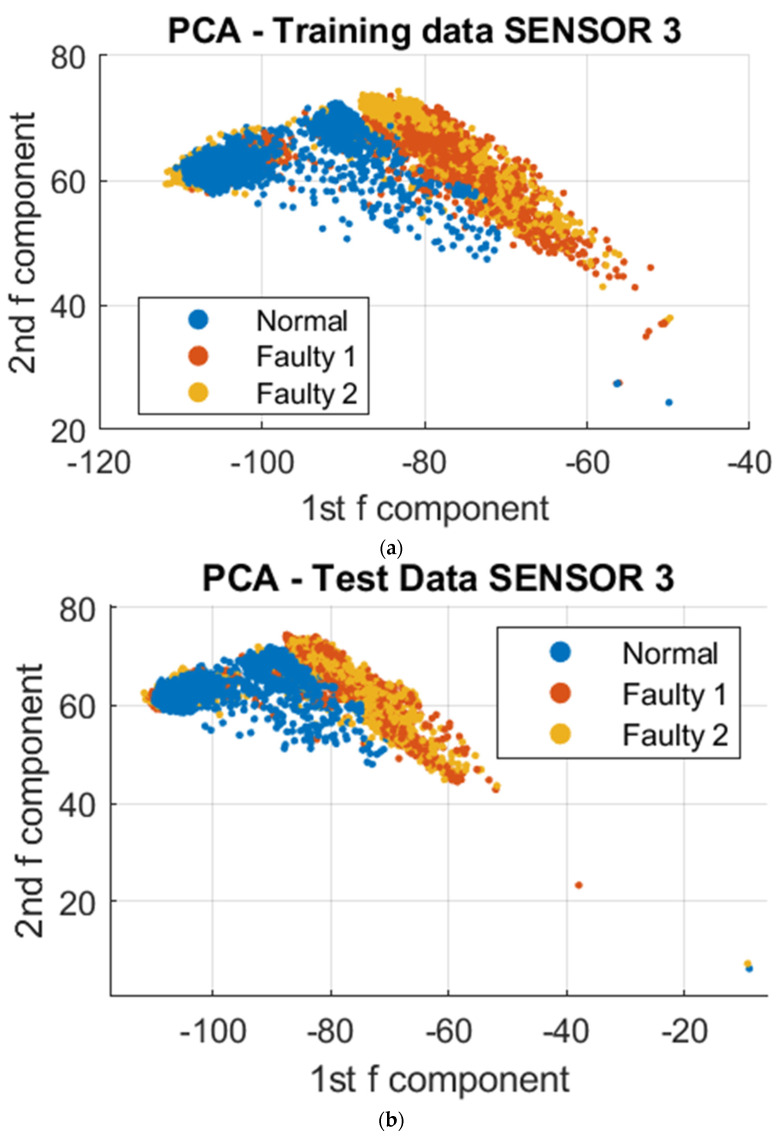
PCA on Sensor 3 at different print temperatures; Faulty 1 and 2 are the unacceptable conditions of 175° and 165°, respectively. (**a**) 2D PCA for trained data; (**b**) 2D PCA for test data; (**c**) 3D PCA for trained data; (**d**) 3D PCA for test data.

**Figure 11 sensors-23-07524-f011:**
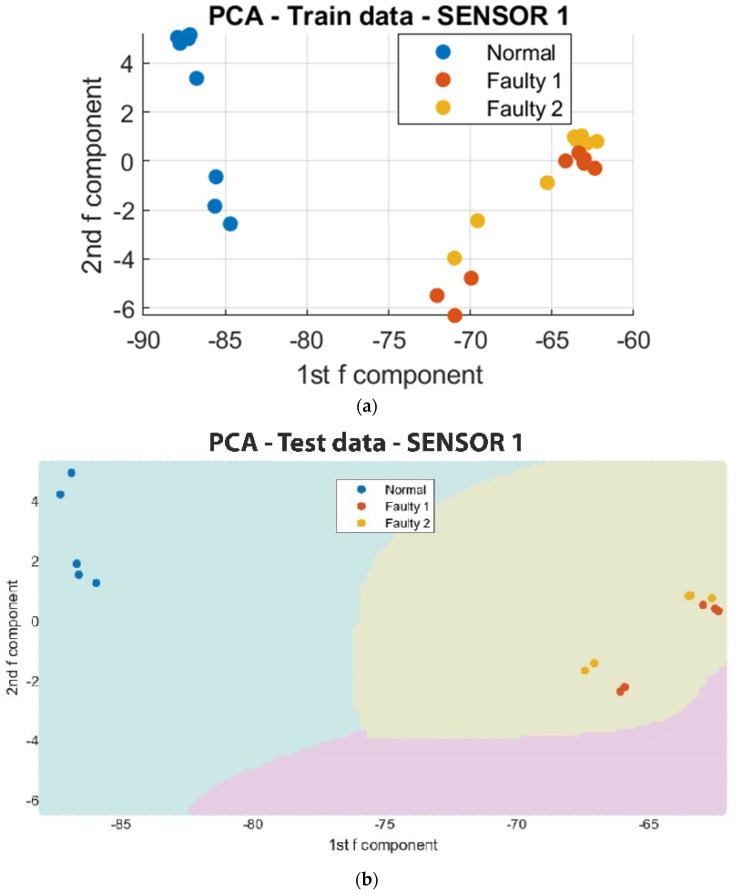
Segmented PCA and SVM for Sensor 1. (**a**) The segmented PCA of the trained data for Sensor 1; (**b**) the segmented PCA and SVM of the test data for Sensor 1.

**Figure 12 sensors-23-07524-f012:**
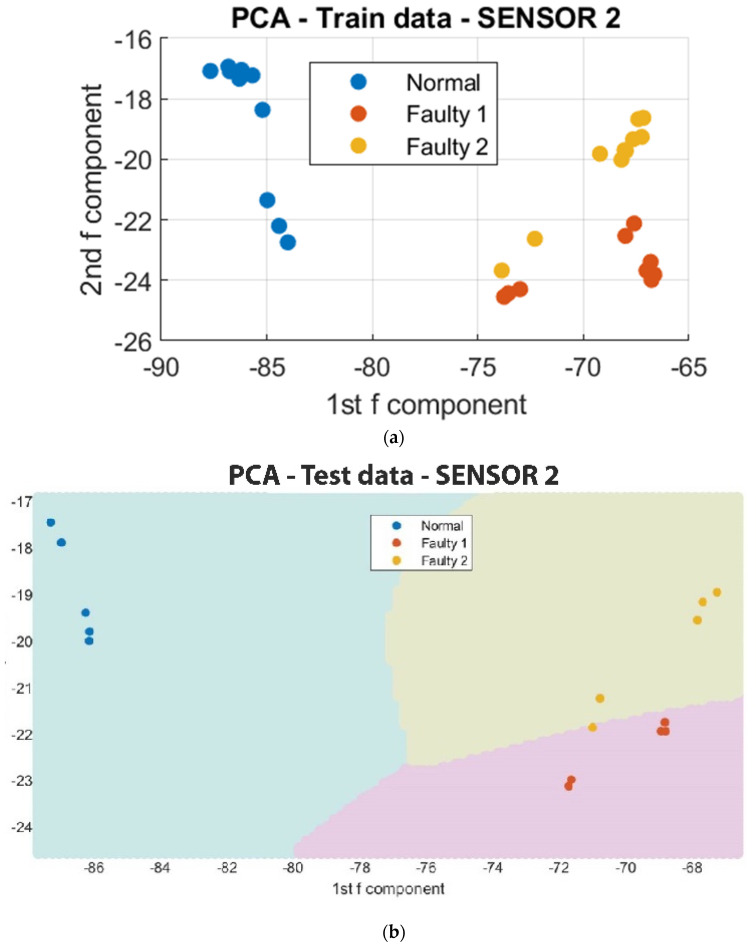
Segmented PCA and SVM for Sensor 2. (**a**) The segmented PCA of the trained data for Sensor 2; (**b**) the segmented PCA and SVM of the test data for Sensor 2.

**Figure 13 sensors-23-07524-f013:**
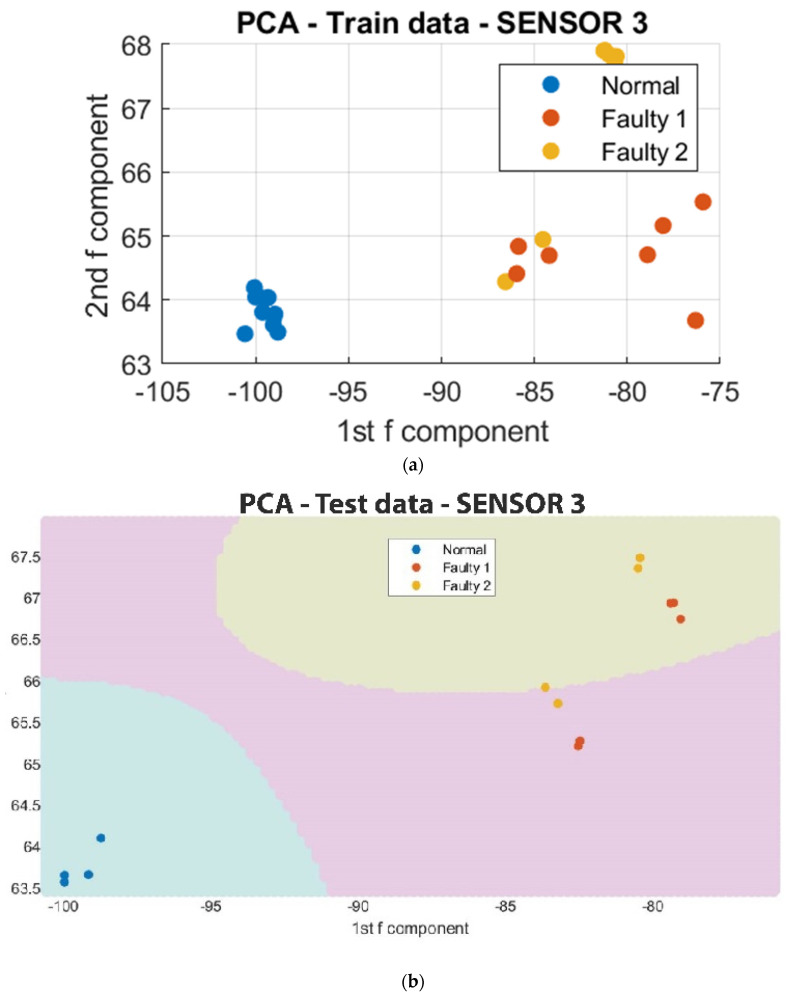
Segmented PCA and SVM for Sensor 3. (**a**) The segmented PCA of the trained data for Sensor 3; (**b**) the segmented PCA and SVM of the test data for Sensor 3.

**Figure 14 sensors-23-07524-f014:**
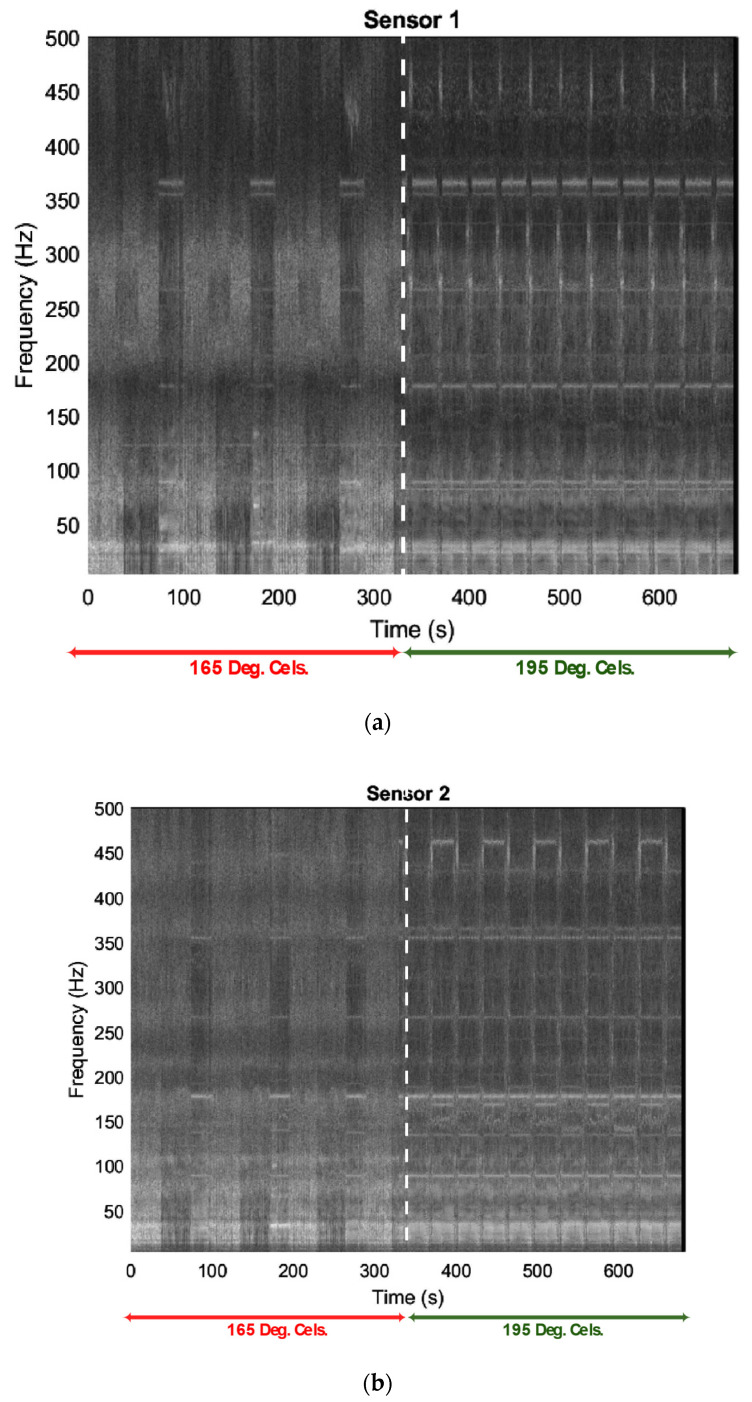
Comparison spectrogram graph at different printing temperatures (165° and 195 °) (**a**) for Sensor 1; (**b**) for Sensor 2; (**c**) for Sensor 3.

**Figure 15 sensors-23-07524-f015:**
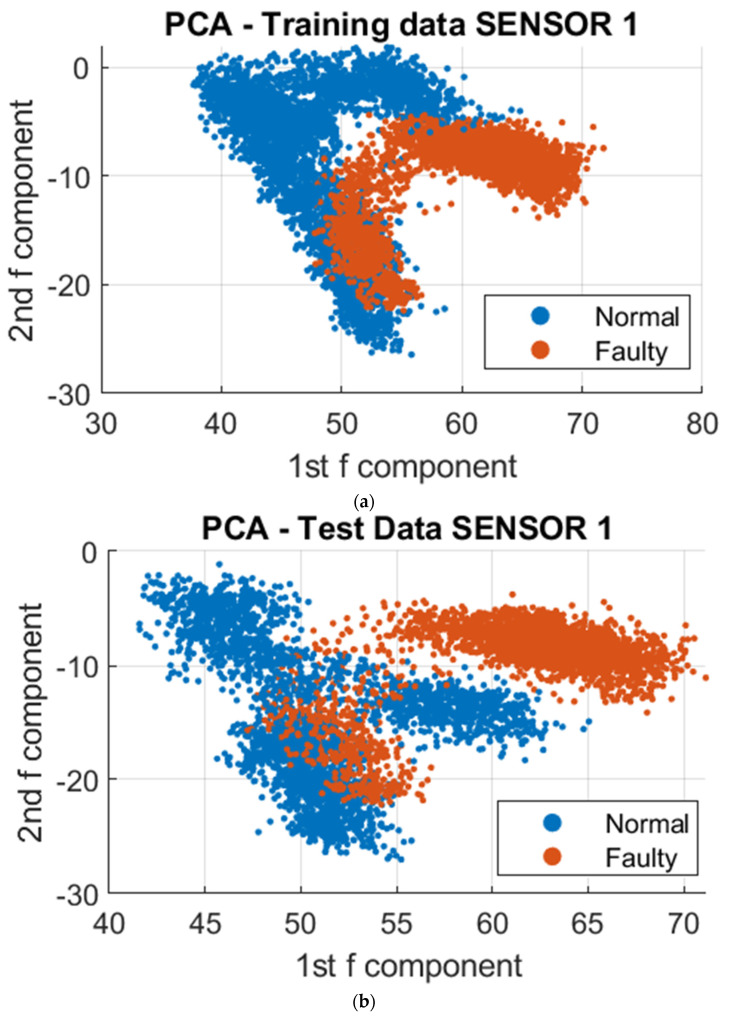
PCA on Sensor 1 at different print temperatures with normal and unacceptable conditions at 195-deg and 165-deg, respectively. (**a**) 2D PCA for trained data; (**b**) 2D PCA for test data; (**c**) 2D segmented PCA for trained data; (**d**) 2D segmented PCA and SVM for test data.

**Figure 16 sensors-23-07524-f016:**
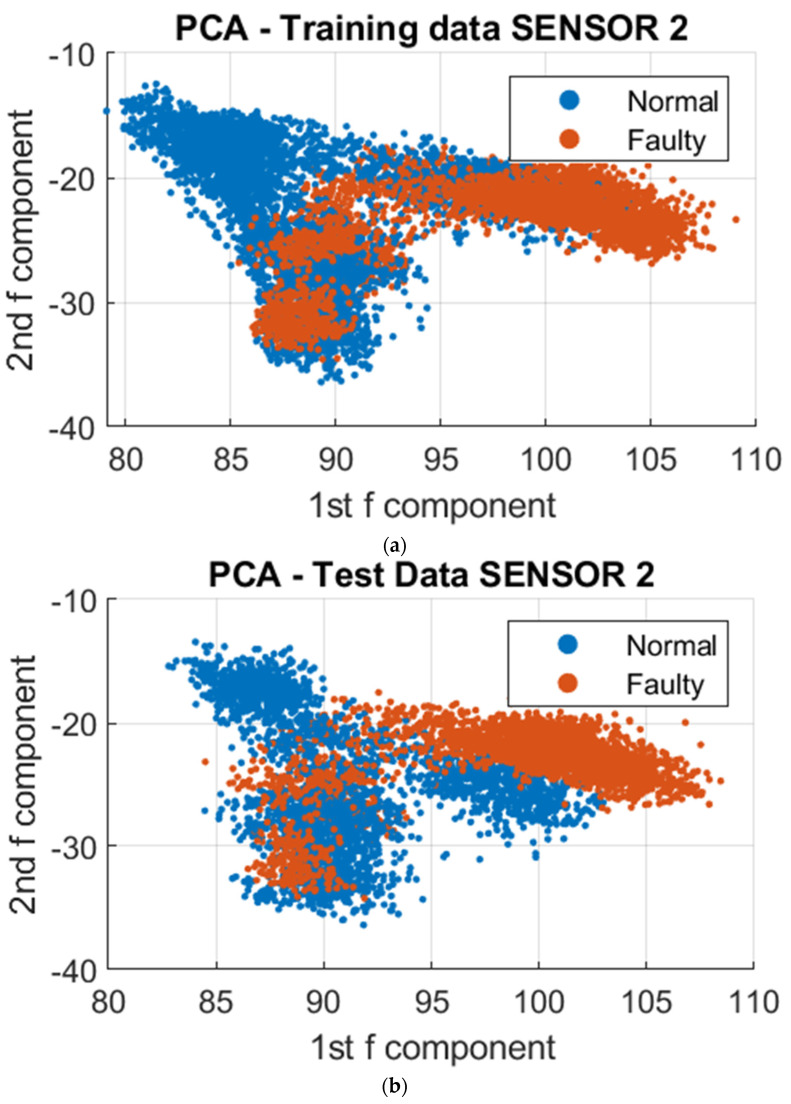
PCA on Sensor 2 at different print temperatures with normal and unacceptable conditions at 195-deg and 165-deg, respectively. (**a**) 2D PCA for trained data; (**b**) 2D PCA for test data; (**c**) 2D segmented PCA for trained data; (**d**) 2D segmented PCA and SVM for test data.

**Figure 17 sensors-23-07524-f017:**
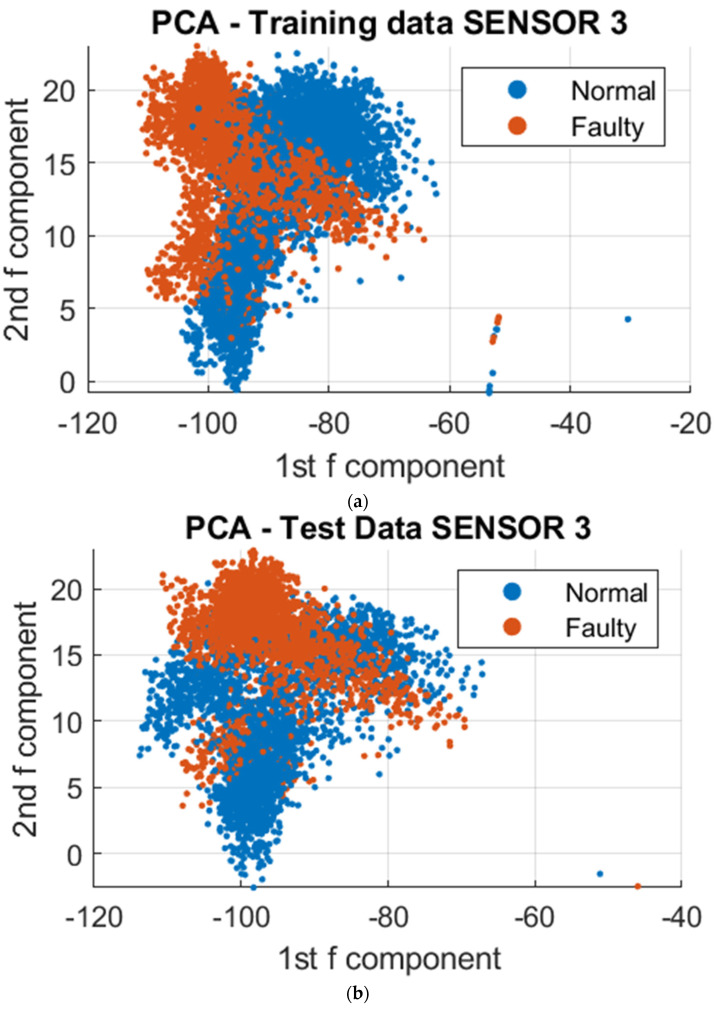
PCA on Sensor 3 at different print temperatures with normal and unacceptable conditions at 195-deg and 165-deg, respectively. (**a**) 2D PCA for trained data; (**b**) 2D PCA for test data; (**c**) 2D segmented PCA for trained data; (**d**) 2D segmented PCA and SVM for test data.

**Table 1 sensors-23-07524-t001:** The set of test conditions.

Printer Function	Nozzle Condition	Schematic of Nozzle Conditions	Extrusion Temp.	Bed Temp.	Print Speed
Normal	Unclogged	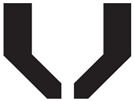	195 °C	60 °C	60 mm/s
Limited	Slight clogging	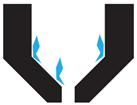	185 °C	60 °C	60 mm/s
Unacceptable	Partially clogged	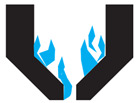	175 °C	60 °C	60 mm/s
Unacceptable	Completely clogged	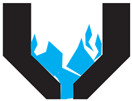	165 °C	60 °C	60 mm/s

**Table 2 sensors-23-07524-t002:** Dominant frequencies at different print states detected using the three sensors.

Sensor	Observed Frequencies			
	Normal	Limited	Unacceptable	Unacceptable
Sensor 1	370 Hz, 270 Hz, 250 Hz	370 Hz, 270 Hz, 250 Hz	370 Hz, 350 Hz, 270 Hz, 180 Hz	370 Hz, 350 Hz, 270 Hz, 180 Hz
Sensor 2	390 Hz, 370 Hz, 250 Hz, 200 Hz	370 Hz, 250 Hz, 220 Hz, 200 Hz	450 Hz, 350 Hz, 270 Hz, 180 Hz	350 Hz, 270 Hz, 180 Hz
Sensor 3	370 Hz, 250 Hz	370 Hz, 250 Hz	350 Hz, 270 Hz, 180 Hz	350 Hz, 270 Hz, 180 Hz

**Table 3 sensors-23-07524-t003:** Dominant frequencies at different print states detected using the three sensors while printing the octopus.

Sensor	Observed Frequencies	
	Normal	Unacceptable
Sensor 1	370 Hz, 270 Hz, 180 Hz	Undiscovered
Sensor 2	370 Hz, 270 Hz, 180 Hz	Undiscovered
Sensor 3	370 Hz, 270 Hz, 180 Hz	Undiscovered

## Data Availability

The data are available in a publicly accessible repository. The data presented in this study are openly available by request.

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
