# Peer review of "Fault Detection in 3D Printing: A Study on Sensor Positioning and Vibrational Patterns"

_sensors, 2023, doi:10.3390/s23177524_

Round 1
Reviewer 1 Report
This work investigates the potential application of accelerometers in identifying vibration patterns in 3D printers that can effectively predict the operating state of the 3D printer. The authors also used advanced data analysis techniques, such as fast Fourier transforms and some machine learning models. And it is investigated that the placement of accelerometer is a key factor for vibration-based fault detection. Before that, I have some questions:
1. The text format of the images in the text is not uniform, and similar images should be stitched together to facilitate viewingï¼›
2. What is the purpose of using both 2D and 3D in Figures 11, 12 and 13ï¼›
3. What method is used for the clustering process does not seem to be specifiedï¼›
4. In line 532, the author states that when the frequency is 0-50 HZ, sensor 1 can distinguish between frequencies at different temperatures, while at 270 HZ, sensors 2 and 3 cannot fail to show the difference. This comparison is incorrect because the frequencies compared are not the same, and the frequencies that the three sensors can adequately detect are between 0-100HZ.
5. In line 526,“Utilizing FFT, the changes in vibrational frequencies were captured by the three sensors, with a particular focus on frequencies between 0 and 50Hz.”How the frequencies in the following paragraphs in the article are displayed?
6. In line 179, Figures 4, 5 and 6, and In line 229, Figure 9 need to be repositioned to facilitate comparison.
7. Figures 14, 15, 16 are not labeled (a), (b), (c).
8. The data in Figure 17 is incorrectly labeled.
.
Author Response
1.The text format of the images in the text is not uniform, and similar
images should be stitched together to facilitate viewing
Answer: The text format of the images is uniform and similar images are
stitched together.
2. What is the purpose of using both 2D and 3D in Figures 11, 12 and 13
Answer: The significance of 3D figures lies in their ability to visually
represent the absence of overlapping clusters across different print
states. While their importance may be perceived as nonessential, they
provide a clear depiction of non-overlapping clustering patterns during
different stages of printing
3. What method is used for the clustering process does not seem to
be specified
Answer: The clustering method employed in this study was Principal
Component Analysis (PCA), which is specified immediately following the
spectrogram analysis.
4. In line 532, the author states that when the frequency is 0-50 HZ, sensor
1 can distinguish between frequencies at different temperatures, while at
270 HZ, sensors 2 and 3 cannot fail to show the difference. This comparison
is incorrect because the frequencies compared are not the same, and the
frequencies that the three sensors can adequately detect are between
0-100HZ.
Answer: The conclusion at line 532 has been updated to reflect the
comparison better.
5. In line 526,“Utilizing FFT, the changes in vibrational frequencies were
captured by the three sensors, with a particular focus on frequencies
between 0 and 50Hz.”How the frequencies in the following paragraphs in the
article are displayed?
Answer: Figure 6 shows that there were varying frequencies observed during
the transition of the nozzle from an unclogged to a completely clogged
condition in the FFT. We have also reworded line 526 to reflect our
observations better.
6. In line 179, Figures 4, 5 and 6, and In line 229, Figure 9 need to
be repositioned to facilitate comparison.
Answer: Figures 4, 5, and 6 have been combined and labeled as 4a, 4b, and
4c. Additionally, Figure 9, previously listed as Figure 6, has been
repositioned.
7. Figures 14, 15, 16 are not labeled (a), (b), (c).
Answer: The corrections have been made.
8. The data in Figure 17 is incorrectly labeled.
Answer: The correction has been made
*also, we are in the process of updating the authorship (adding Hieu Nguyen) so the authors on the attached pdf do not all appear on MDPI yet.
** We have added a highlighted manuscript with the changes suggested by all three reviewers. This should make it easier to follow the revisions.

Reviewer 2 Report
- Point 4 should be Conclusion only. In conclusion, pls don't start like This paper presents a........it gives a feeling of an 'Abstract'
rather write like
'This paper has presented.........'
-Ref [8] is incorrectly detailed, the correct details are:
Durwesh Jhodkar, Ankit Nayak, Kapil Gupta (2021), “Experimental investigation of nozzle clogging using vibration signal-based condition monitoring for fused deposition modeling”, Materials Science Forum (TransTech Publication)Vol. 1037, pp 55-64
-Figs 1-3 are very poor quality figs. Fig 1- the naming background white is not appearing nice.
-Some quantitative results are also required to be briefed in the Abstract.
I find the English fine.
Author Response
1. Point 4 should be Conclusion only. In conclusion, pls don't start like
This paper presents a........it gives a feeling of an 'Abstract'rather
write like'This paper has presented.........'
Answer: We have corrected the conclusion.
2. Ref [8] is incorrectly detailed, the correct details are:Durwesh Jhodkar,
Ankit Nayak, Kapil Gupta (2021), “Experimental investigation of nozzle
clogging using vibration signal-based condition monitoring for fused
deposition modeling”, Materials Science Forum (TransTech Publication)Vol.
1037, pp 55-64
Answer: The reference has been updated.
3. Figs 1-3 are very poor quality figs. Fig 1- the naming background white is
not appearing nice.
Answer: We have made improvements to the resolution and clarity of all the
figures. The resolution has been increased to 300 DPI, resulting in better
visuals.
Also, figure 1 has been reworked on for better contrast.
4. Some quantitative results are also required to be briefed in the
Abstract.
Answer: We have included a brief quantitative result to the Abstract
*Also, we are in the process of updating the authorship of the paper to include Hieu Nguyen so the authors on the attached pdf may not match the authors in MDPI.
** We have added a highlighted manuscript with the changes suggested by all three reviewers. This should make it easier to follow the revisions.

Reviewer 3 Report
see attached pdf

Author Response
We have included the suggested articles and also have up to 40
references. This article discusses how vibrational patterns can detect
nozzle clogs and wobbling of detached samples. Our future research will
explore the potential of using vibrational patterns in quality control to
identify z-banding and skewness during fabrication.

Round 2
Reviewer 3 Report
The authors practically did not upload a point-to-point answer to my review comments. Their "author response.pdf" is the same file as the revised manuscript. I tried to find the differences in the revised manuscript, but it is very difficult.
Authors seem to ignore basic author obligations and procedures when writing a manuscript for a scientific journal. I suggest its rejection.
Quality of English Language is fine.
Author Response
|
Responses to Reviewer 3 |
||
|
Comments |
Responses |
|
|
1 |
It would bе bеnеficial if thе articlе dеlvеd dееpеr into thе spеcific typеs of faults or opеrational statеs that can bе dеtеctеd using vibrational pattеrns. |
The authors are grateful to the reviewer for going through the manuscript in-depth. The current manuscript focuses on vibrational patterns observed as a result of nozzle conditions in line 175. This is a continuous study; our future research will focus on using vibrational patterns to predict z-banding and skewness effects in parts during fabrication, as captured in line 825.
|
|
2 |
Additionally, providing morе dеtails on thе еvaluation mеtrics usеd to assеss thе pеrformancе of thе machinе lеarning modеls would еnhancе thе comprеhеnsibility of thе study. |
The scope of our manuscript wasn’t on machine learning; rather, we leveraged the widely known PCA and SVM to clearly show the differences in the vibration patterns observed in FFT and Spectrogram. However, in reviewing the manuscript, we have enhanced the comprehensibility of the study. The reviewer’s comments have added to the overall quality of our manuscript. |
|
3 |
Authors should also expand the references list which is very small. The following references should be added in lines 33-35. References list, should include at least 40 references, so authors are encouraged to add even more.
10.3844/ajeassp.2023.12.22 10.3390/app13084777 10.1016/j.promfg.2019.06.089 |
We have included and referenced the suggested articles in lines 873, 882 and 891, respectively. We have also added more references. |
Also we have added a highlighted manuscript that should make it easier to track the various revisions to the manuscript. The highlights represent changes suggested by all three reviewers.
Round 3
Reviewer 3 Report
The corrections are visible now, and the article can be successfully reviewed. The authors have made adequate revisions and I believe, now, that the manuscript is ready for publication.